The influence of sample distribution on growth model output for a highly-exploited marine fish, the Gulf Corvina (Cynoscion othonopterus)

Bolser Derek G. 1
Grüss Arnaud 2
Lopez Mark A. 1
Reed Erin M. 1 3
Mascareñas-Osorio Ismael 4
Erisman Brad E. berisman@utexas.edu 1
1 Marine Science Institute, University of Texas at Austin , Port Aransas , TX , United States of America
2 School of Aquatic and Fishery Sciences, University of Washington , Seattle , WA , United States of America
3 Joint Institute for Marine and Atmospheric Research, University of Hawaii and NOAA Fisheries, Pacific Islands Fisheries Science Center , Honolulu , HI , United States of America
4 Centro para la Biodiversidad Marina y la Conservación , La Paz , Mexico
Esteban María Ángeles
Electronic publication date: 2018 Sep 17
Publication date: 2018
Volume: 6
Electronic Location ID: e5582
Received 2018 Apr 13; Accepted 2018 Aug 13
Copyright: ©2018 Bolser et al.
Copyright year: 2018
Copyright holder: Bolser et al.
License: This is an open access article distributed under the terms of the Creative Commons Attribution License, which permits unrestricted use, distribution, reproduction and adaptation in any medium and for any purpose provided that it is properly attributed. For attribution, the original author(s), title, publication source (PeerJ) and either DOI or URL of the article must be cited.
License URL: https://creativecommons.org/licenses/by/4.0/

Keywords: Gulf of California, Gulf Corvina, Growth modelling, Fish growth, Data-poor fisheries, Highly-exploited, Vulnerable

Funding: Walton Family Foundation World Wildlife Fund Mexico The Environmental Defense Fund This work was supported by grants from The Walton Family Foundation, World Wildlife Fund Mexico, and the Environmental Defense Fund. The funders had no role in study design, data collection and analysis, decision to publish, or preparation of the manuscript.

==============================
Estimating the growth of fishes is critical to understanding their life history and conducting fisheries assessments. It is imperative to sufficiently sample each size and age class of fishes to construct models that accurately reflect biological growth patterns, but this may be a challenging endeavor for highly-exploited species in which older fish are rare. Here, we use the Gulf Corvina (Cynoscion othonopterus), a vulnerable marine fish that has been persistently overfished for two decades, as a model species to compare the performance of several growth models. We fit the von Bertalanffy, Gompertz, logistic, Schnute, and Schnute–Richards growth models to length-at-age data by nonlinear least squares regression and used simple indicators to reveal biased data and ensure our results were biologically feasible. We then explored the consequences of selecting a biased growth model with a per-recruit model that estimated female spawning-stock-biomass-per-recruit and yield-per-recruit. Based on statistics alone, we found that the Schnute–Richards model described our data best. However, it was evident that our data were biased by a bimodal distribution of samples and underrepresentation of large, old individuals, and we found the Schnute–Richards model output to be biologically implausible. By simulating an equal distribution of samples across all age classes, we found that sample distribution distinctly influenced model output for all growth models tested. Consequently, we determined that the growth pattern of the Gulf Corvina was best described by the von Bertalanffy growth model, which was the most robust to biased data, comparable across studies, and statistically comparable to the Schnute–Richards model. Growth model selection had important consequences for assessment, as the per-recruit model employing the Schnute–Richards model fit to raw data predicted the stock to be in a much healthier state than per-recruit models employing other growth models. Our results serve as a reminder of the importance of complete sampling of all size and age classes when possible and transparent identification of biased data when complete sampling is not possible.

Introduction

Age and size data inform estimates of life history parameters that are crucial to fisheries stock assessments. In early assessments such as Beverton and Holt’s yield-per-recruit model (1957), size at age was critical for estimating reproductive output and thus the sustainability of fisheries. In today’s age-structured stock assessments, size at age is used to convert from biomass to number of fish, determine selectivity, and calculate expected length compositions (Francis, 2016). Similarly, size (i.e., length or weight) at age is used in size-structured stock assessment models to inform transitions between size bins and determine length composition (Punt, Haddon & McGarvey, 2016). Accurately representing the relationship between size and age is particularly important for vulnerable fish and in data-poor fisheries, in which life-history parameters and population structure often drive stock assessments and management decisions (Dulvy et al., 2004; Froese, 2004; Honey, Moxley & Fujita, 2010; Hordyk et al., 2016). Specifically, these types of assessments rely heavily on age–length data to confer insights on vulnerability and overfishing (Erisman  et al.,  2014).

When modelling the relationship between age and size for the purposes of assessment, and for any purpose, each age and size class must be sufficiently represented to generate growth parameter estimates that reflect biological growth (Cailliet et al., 1986; Cailliet & Tanaka, 1990; Francis & Francis, 1992; Cailliet & Goldman, 2004). It is important to make the distinction between this type of sampling and sampling to reflect population structure, which should not be the goal of age and growth studies as this reflects bias due to the relative scarcity of large and old individuals. Sufficiently representing each size and age class may be especially difficult in highly-exploited species, as exploitation alters the population structure of fishes by preferentially selecting for large and old fish individuals (Mason, 1998; Berkeley et al., 2004). The ramifications for failing to acknowledge selection are clear, as length-selective fishing mortality distorts growth curves (Walker et al., 1998). Further, the lack of representation of large and old individuals could result in underestimation of lifespan and longevity, which makes fishery management measures less effective (Campana, 2001; Cailliet & Andrews, 2008; Goldman et al., 2012). Large and old fish drive estimates of the maximum average length parameter L∞, and without them, L∞ is underestimated and the growth rate (typically denoted by K) is overestimated. The underestimation of L∞ and the overestimation of K lead to the assumptions of a shorter generation time and less mortality, and thus more resiliency to high levels of fishing pressure (Campana, 2001; Goldman et al., 2012; Harry, 2017). The L∞ term is particularly important when growth models are incorporated into stock assessment (Wells et al., 2013). This problem may also occur in growth modelling for vulnerable fish or in data-poor fisheries, where lack of representation of each age and size class due to sampling constraints or the scarcity of individuals may similarly affect parameter estimates. Fishery dependent data are often the only data available for growth modelling, which may be acceptable only as long as the inherent biases and limitations are acknowledged.

Several models have been developed to quantify the relationship between age and size, with body length being the most common metric of size. Typically, asymptotic growth models are used to quantify this relationship. These models describe fast growth in the earliest years of life and slower growth in later years. Despite some criticism (Roff, 1980; Czarnołe‘ski & Kozłowski, 1998), the most widely used is the von Bertalanffy growth model (Chen, Jackson & Harvey, 1992; Kimura, 2008). Rooted in bioenergetics, this model is intended to give a biologically relevant representation of how catabolic and anabolic processes work within a fish to change growth over the lifespan of fishes (Von Bertalanffy, 1938; Pauly, 2010). Over the years, there have been many re-parameterizations of the von Bertalanffy model with incorporation of growth-influencing factors and applications to a variety of situations (Gallucci II & Quinn, 1979; Ratkowsky, 1986; Helser & Lai, 2004; Kimura, 2008; Brunel & Dickey-Collas, 2010; Van Poorten & Walters, 2016), but the original parametrization is still the most commonly used (Lorenzen, 2016). Other asymptotic growth models are commonly used in fisheries, such as the Gompertz growth model (Gompertz, 1825) and the logistic growth model (Ricker, 1975).

In recent years, fish growth models have moved from a foundation in bioenergetics to being more statistically driven (Van Poorten & Walters, 2016). These models are inherently more flexible, allowing them to capture subtleties in growth patterns that may not be captured by the more inflexible growth models. The Schnute model (Schnute, 1981), for example, has four curve families that the model may assume based on which types of data the model is fit to and what other functions are incorporated into the framework. Another flexible growth model, the Schnute–Richards model (Schnute & Richards, 1990), can describe biphasic growth among several other forms. By design, the Schnute–Richards model may be equivalent to the other growth models discussed above when the proper values are specified for its dimensionless parameters. Fish growth is inherently plastic and fish do not all grow the same way (Weatherley, 1990; Lorenzen, 2016), so a flexible growth model may be advantageous in certain situations. However, these flexible models may also be more sensitive to sampling biases in data, potentially producing growth patterns that reflect the size-frequency distribution of fish collected rather than the biological growth pattern of the species.

The Gulf Corvina (Cynoscion othonopterus) is an ideal species to examine the performance of multiple growth models in a highly-exploited marine fish. Endemic to the northern Gulf of California, Mexico (Robertson & Allen, 2008), it is currently listed as vulnerable under the International Union for the Conservation of Nature (IUCN) Redlist (Chao et al., 2010). Gulf Corvina have experienced persistent overfishing on their spawning aggregations for the past two decades, which have resulted in growing concern for the fishery’s stability and longevity (Erisman et al., 2012; Ruelas-Peña, Valdez-Muñoz & Aragón-Noriega, 2013; Erisman et al., 2014; Ortiz et al., 2016). The life history of this species has been well documented and provides an ideal foundation for future analysis of individual and population growth (Román-Rodríguez, 2000; Gherard et al., 2013). With a documented maximum size of 1,013 mm total length (TL) and a documented maximum age of 9 years, Gulf Corvina is a fast-growing, short-lived species which reaches sexual maturity at 2 years of age (Gherard et al., 2013). However, the combination of highly efficient, size-selective gear and persistent overfishing have severely truncated the age structure of the population (Erisman et al., 2014; Ortiz et al., 2016). The mean age of capture of Gulf Corvina is 5 years (ca. 700 mm TL), and few individuals older than age 7 or younger than age 4 have been observed in the fishery (Gherard et al., 2013; Erisman et al., 2014; Ortiz et al., 2016).

Past studies of Gulf Corvina growth, which have relied solely on fishery-dependent data with incomplete sampling of all size and age classes, have produced different results due to differences in model selection approach. Based on the congruence of the model with the growth pattern of many species of the genus Cynoscion and other sciaenid fishes (Rutherford, Thue & Buker, 1982; Lowerre-Barbieri, Chittenden & Barbieri, 1995; Rodriguez & Hammann, 1997), Gherard et al. (2013) took a conservative, single model approach and fit the von Bertalanffy growth model to Gulf Corvina age–length data. Conversely, Aragon-Noriega (2014) chose a statistically-driven approach and fit several models to multiple datasets, concluding that Gulf Corvina grew in a biphasic pattern with slow growth in the beginning of life, rapid growth after age two, and slow growth after age four. Notably, Aragon-Noriega’s (2014) estimates for the L∞ parameter varied greatly, from 735.0 to 1,126.6 mm, depending on which dataset was used. Given this variability, absence of biphasic growth patterns in similar sciaenids, and the distance from the maximum observed length of Gulf Corvina (1,013 mm; Gherard et al., 2013), Aragon-Noriega’s (2014) estimates may be biologically unrealistic. Mendivil-Mendoza et al. (2017) took a similar approach and found a similarly wide range of L∞ values (666.7–1,306.0 mm). However, despite fitting models to similar data as Aragon-Noriega (2014), Mendivil-Mendoza et al. (2017) did not describe the biphasic growth pattern recorded by Aragon-Noriega (2014). The existence of discrepancies between the previous Gulf Corvina growth studies and the importance of the age–length relationship to the stock assessment of the fishery merit further investigation on the growth pattern of the species.

Here, we model the growth of Gulf Corvina and draw conclusions about data needs and fisheries assessments. Our specific objectives were to: (1) assess how representation of size and age classes affected growth parameter estimates and (2) compare the performance of multiple growth models for describing age-at-length data for Gulf Corvina. Through generating a more complete dataset than previous studies and testing for biases in our data with simple indicators, we addressed these objectives. Moreover, using the results of simulations with a per-recruit model, we discussed the implications of misrepresenting growth in highly-exploited, vulnerable marine fishes.

Materials and Methods

Data collection

Seven hundred and forty-nine Gulf Corvina were sampled from 2009 through 2013 at the three locations in the upper Gulf of California: El Golfo de Santa Clara (Sonora), San Felipe (Baja California), and El Zanjón (Baja California). Information on total length (TL) was recorded to the nearest mm for each fish collected, and the sagittal otoliths were removed, dried whole and stored until further use. Five hundred and thirty of these samples were collected from the commercial Gulf Corvina fishery and from bycatch from the shrimp fishery. These data were used by Gherard et al. (2013). In order to increase representation of size and age classes that were scarce in the dataset used by Gherard et al. (2013), we collected 219 additional samples in 2012–2013 from the bycatch of other fisheries (e.g., shrimp), fishery-independent sampling of small individuals (<30 cm TL), and the commercial Gulf Corvina fishery. All fish were deceased at the time of collection from fishers. The research protocol was approved under UCSD IACUC ID no. S13240, and data were collected under CONANP permit no. CNANP-00-007.

Otolith preparation and ageing protocols were followed according to the methods developed by Gherard et al. (2013) for Gulf Corvina. Whole sagittal otoliths were first mounted on wood blocks with a cyanoacrylate adhesive and a 0.5 mm dorsal-ventral cross-section was cut through the otolith focus using a double-bladed Buehler Isomet 1000 precision saw (Allen et al., 1995). Sub-sections were then mounted on a glass slide using thermoplastic glue and submerged in a glass petri dish with water and a black background. Transmitted light under a Zeiss Stemi 2000-C microscope with a Zeiss Axiocam 105 color camera at 6.25× total magnification was used to count the alternating opaque and translucent growth zones that define an annulus (Fig. 1). For the purposes of this study, an annulus was defined as one full opaque and translucent zone of growth (Cailliet et al., 1996), which was validated for Gulf Corvina by previous studies (Román-Rodríguez, 2000; Rowell et al., 2005; Gherard et al., 2013). Each otolith was aged by two independent readers from digital images of cross-sections, as direct observation through the scope did not distort band pattern and did not affect age estimates. Samples were excluded from analysis when discrepancies between readers occurred.

Figure 1 Transverse section of a sagittal otolith from a five-year old Gulf Corvina.

Annuli are numbered and marked by white rectangles. Transmitted light under a Zeiss Stemi 2000-C microscope with a Zeiss Axiocam 105 color camera at 6.25× total magnification was used to count the alternating opaque and translucent growth zones that define an annulus.

Model fitting and assessment of fit

Growth modelling

A suite of growth models was fit to age data determined from otoliths as described, and length data obtained in the field. Model parameters were estimated using non-linear least squares regression with the Levenberg–Marquardt algorithm, and confidence limits were placed around parameter estimates in R studio (using the R packages Ogle, 2017; Elzhov et al., 2015 and Baty et al., 2015).

The specialized von Bertalanffy growth model (Von Bertalanffy, 1938) is given by: (1) Lt=L∞1−e−Kt−t0

where L(t) is size (in mm TL) at age t, L∞ is the maximum average length (in mm TL), K is the growth rate coefficient (in year−1), and t 0 is the theoretical age at which length is zero (in years).

The Gompertz growth model (Gompertz, 1825) is given by: (2) Lt=L∞e−1Ke−Kt−t0

where the consistent parameters are the same as described for Eq. (1), K1 is the rate of exponential decrease of the relative growth rate with age (with units of yr−1), and t1 is 1∕k4 ln λ, where λ is the theoretical initial relative growth rate at zero age (with units of yr−1).

The logistic model (Ricker, 1975) is given by: (3) Lt=L∞1+e−Kt−t0−1

where the consistent parameters are the same as described for Eqs. (1) and (2), K2 is a relative growth rate parameter (with units of yr−1), and t2 is the inflection point of the curve.

The Schnute model where a and b are not equal to zero (Schnute, 1981) is given by: (4) Lt=L1b+L2b−L1b1−e−at−T11−e−aT2−T11∕b

where T1 is the first specified age, T2 is the second specified age, L 1 is size at age T1, L2 is size at age T2, a is the constant relative rate of relative growth (in year−1), and b is the incremental relative rate of relative growth (dimensionless),

Finally, the Schnute–Richards model (Schnute & Richards, 1990) is given by: (5) Lt=L∞1+αe−atc1∕b

where α, b, and c are dimensionless parameters, and a has the unit of year−b.

Statistical measures of fit

Model fit was assessed with the bias-corrected Akaike Information Criterion (AICc) (Shono, 2000; Burnham & Anderson, 2004), and Bayesian Information Criterion (Schwarz, 1978) in R Studio (using the R package (Mazerolle, 2017)).

The formula for AICc is given by: (6) AICc=AIC+2kk+1n−k−1

where: (7) AIC=−2logL+2k

and n is the number of observations, k is the number of model parameters, and L is the likelihood.

The formula for BIC is given by: (8) BIC=2lnL+klogn

where parameter definitions are the same as described for Eq. (7).

The smallest AICc and BIC values indicate the best model. The difference between the two criteria is that AICc is designed to select the model that describes reality the best while treating no models as true, which is consistent with an information theory approach, whereas BIC is designed to select the true model. Practically, BIC penalizes for the number of parameters more heavily than AICc. AICc was used instead of AIC as it is bias-corrected at small n values or high k:n ratios; AICc converges to AIC at large n values (Burnham & Anderson, 2004). AICc and BIC values were calculated to show the absolute difference between model fits. Next, AICc weights were calculated for model averaging of parameter estimates; the AIC weighting formula is given by: (9) wi=e−0.5Δi∑k=15e−0.5Δk

where parameter definitions are the same as described for Eqs. (7) and (8).

Simple indicators of biased data

Simulation of an ideal sampling outcome

To test for the influence of sampled population structure on growth model output, different amounts of simulated data were added to raw data so that each age observed (1–8) had 200 total observations. Data were simulated from a normal distribution with the same mean and standard deviation as the raw data at each age class. This simulation was not intended to generate the true population structure of Gulf Corvina in the Gulf of California, but rather to generate an equal number of samples in each age and size class. This simulation did not explicitly account for selectivity or limits in sampling effort, but filled in gaps left by these factors and others that prevented more equal representation of each size and age class in the raw data. Models were fit to the new dataset and goodness of fit was assessed in the same manner as was described above.

Froese and Binohlan’s empirical relationship

Froese & Binohlan’s (2000) empirical relationship between the longest fish in the data set (Lmax) and L∞ was used to specifically test for the influence of the lack of large and old fish in the raw dataset, which is likely due to heavy exploitation. If large and old fish are insufficiently represented in the dataset, it stands to reason that the L∞ predicted by this relationship will be greater than the modelled L∞. This relationship is given by: (10) logL∞=0.044+0.9841∗ logLmax

Literature review

A brief literature review of sciaenid growth modelling was conducted to assess how the results of this study compared with other studies on fishes closely related to the Gulf Corvina (e.g., other species in the genus Cynoscion). In conjunction with Froese and Binohlan’s empirical relationship and the simple simulation of an ideal sampling scenario, this brief literature review was intended to check if the samples used in this study produced a biologically plausible growth pattern when growth was modelled.

Simulations with a per-recruit model

To be able to discuss the implications of misrepresenting growth in Gulf Corvina, we ran simulations with a per-recruit model detailed in Appendix S1. In brief, this per-recruit model estimates the female spawning-stock-biomass-per-recruit (SSBR; a proxy of reproductive capacity) and yield-per-recruit (YPR; exploitable biomass) of Gulf Corvina in relation to the annual exploitation rates of the old adults (≥5 year-old individuals) of the species (EOA). In this per-recruit model, Gulf Corvina are assumed to grow according to one of five alternative growth models: (1) the von Bertalanffy model developed in Gherard et al. (2013), referred to as the “Gherard model”; (2) the von Bertalanffy model fit to raw data in the present study; (3) the von Bertalanffy model fit to raw data bolstered by simulation values in this study; (4) the Schnute–Richards model fit to raw data in the present study; and (5) the Schnute–Richards model fit to raw data bolstered by simulation values in this study. The current EOA was estimated to be 0.825 year−1 (Appendix S1). We first ran simulations with the per-recruit model to determine the maximum value of the YPR of Gulf Corvina (Y PRmax) and the natural SSBR of Gulf Corvina (NSSBR), i.e., its SSBR in the absence of fishing (Appendix S1). Then, we estimated the current fraction of NSSBR (current FNSSBR, i.e., the ratio of current SSBR to NSSBR) and the current YPR over Y PRmax of Gulf Corvina, when each of the five abovementioned growth models is used to represent the growth in length of Gulf Corvina.

Results

Length and age structure

A bimodal distribution was observed in the length and age structure of the fish used in this study (Figs. 2 and 3). The first mode of the distribution represents Gulf Corvina caught as bycatch, whereas the second represents Gulf Corvina caught in the targeted fishery. Lengths ranged from 141–1,013 mm TL, and ages ranged from 1–8 years.

Figure 2 Total length frequency of Gulf Corvina from raw data represented in 10 mm bins.

A bimodal distribution was observed, with the first consisting of Gulf Corvina caught as bycatch, and the second largely consisting of fish from the directed fishery. Few fish larger than 750 mm are present in this dataset.

Figure 3 Age frequency of Gulf Corvina from raw data.

A bimodal distribution was observed, with the first consisting of Gulf Corvina caught as bycatch, and the second largely consisting of fish from the directed fishery. Few fish older than age 6 are present in this dataset.

Model fitting and assessment of fit for models fit to raw data

Growth patterns and parameter estimates for models fit to raw data

The Gompertz, logistic, and von Bertalanffy models yielded asymptotic growth patterns, while the Schnute–Richards model described biphasic growth and the Schnute model described near-linear growth after 1.5 years of life (Fig. 4). Modelled length at age was most similar among models at intermediate ages, where samples were most abundant (Fig. 4). Conversely, modelled length at age was most variable at young and old ages, where samples were most scarce (Fig. 4). Estimates of L∞ ranged from 730.91 mm (Schnute–Richards model) to 916.05 mm (von Bertalanffy model). All parameter estimates are summarized in Table 1, while confidence intervals around parameter estimates are reported in Appendix S2.

Figure 4 Growth models fit to raw age-length data for Gulf Corvina.

The Gompertz, Logistic, and von Bertalanffy models yielded asymptotic growth patterns. However, the Schnute–Richards model described bi-phasic growth, and the Schnute model describes near-linear growth after the first 1.5 years of life. Differences in modelled size at age were most pronounced at the beginning and end of life, where samples were most scarce.

Table 1 Parameter estimates for growth models fit to raw age-length data for Gulf Corvina.

Estimates of L∞ were variable, but not as variable as those reported in previous multi-model studies of Gulf Corvina growth (Aragon-Noriega, 2014; Mendivil-Mendoza et al., 2017). Confidence intervals around parameter estimates may be found in the Supplemental Information.

Model name	Model equation when fit to raw data	
von Bertalanffy	Lt=916.051−e−0.28t−−0.17	
Gompertz	Lt=820.64e−10.51e−0.51t−1.29	
Logistic	Lt=778.881+e−0.76t−1.92−1	
Schnute	Lt=141−0.33+1013−0.33−141−0.331−e−3.36t−11−e−3.368−−11∕−0.33	
Schnute–Richards	Lt=730.911+−0.003e−0.12t2.181∕0.003	

Measures of statistical fit for models fit to raw data

AICc and BIC values indicated that the Schnute–Richards model described the raw data best, followed by the logistic, Gompertz, von Bertalanffy, and Schnute models (Table 2). The AIC weighting formula gave full support to the Schnute–Richards model, so no model averaging of parameters was necessary.

Table 2 Statistical measures of fit for growth models fit to raw age-length data for Gulf Corvina.

The Schnute–Richards model fit the data best according to AICc and BIC values, but is only marginally better than the logistic, Gompertz, and von Bertalanffy models. Note: K indicates the number of parameters.

Model	K	AICc	ΔAICc	AICc weight	BIC	ΔBIC	
Schnute–Richards	6	8,759.82	0.00	1	8,787.42	0.00	
Logistic	4	8,773.62	13.80	0	8,792.04	4.62	
Gompertz	4	8,789.69	29.87	0	8,808.11	20.69	
von Bertalanffy	4	8,813.66	53.84	0	8,832.08	44.66	
Schnute	3a	9,148.78	388.96	0	9,162.61	375.19	
Notes.

a Three parameters were estimated by nonlinear least squares, but four additional parameters were manually inputted (maximum and minimum ages and lengths) for the Schnute model.

Figure 5 Growth models fit to raw Gulf Corvina age-length data bolstered by simulated values.

All models except for the Schnute described asymptotic growth, and only showed slight differences in modelled size at age. Differences in modelled size at age were most pronounced at the beginning and end of life.

Simple indicators of biased data

Growth patterns and parameter estimates for models fit to raw data bolstered by simulated values

The Schnute–Richards, Gompertz, logistic and von Bertalanffy growth models yielded asymptotic growth patterns, while the Schnute model described near-linear growth after 1.5 years of age (Fig. 5). Modelled length at age was similar at intermediate ages among all growth models except for the Schnute model, but differed slightly at young and old ages (Fig. 5). Estimates of L∞ ranged from 834.34 mm (logistic model) to 951.30 mm (von Bertalanffy model) (Table 3). All parameter estimates for each growth model are summarized in Table  3.

Table 3 Parameter estimates for growth models fit to raw Gulf Corvina age-length data bolstered by simulated values.

Compared to parameter estimates for models fit to raw data alone, estimates of L∞ were less variable and generally increased. These estimates are closer to the L∞ of 1,006 predicted by Froese & Binohlan’s (2000) empirical relationship between L∞ and the longest fish in a dataset.

Model	Model fit to data bolstered by simulated values	
von Bertalanffy	Lt=951.301−e−0.25t−−0.33	
Gompertz	Lt=870.48e−10.62e−0.62t−1.34	
Logistic	Lt=834.341+e−0.62t−2.10−1	
Schnute	Lt=141−0.78+1013−0.78−141−0.781−e−6.06t−11−e−6.068−11∕−0.78	
Schnute–Richards	Lt=938.801+−0.0046e−0.67t0.721∕0.0019	

Measures of statistical fit for models fit to raw data bolstered by simulated values

The von Bertalanffy growth model described the raw data bolstered by simulated values best according to AICc and BIC values (Table 4). However, it was only marginally better than the Schnute–Richards and Gompertz models based on AICc. Thus, the von Bertalanffy growth model received 53% of AICc weighting to the Schnute–Richards’ 33%, and Gompertz’ 15%. The logistic model fit the data better than the Schnute model, but neither models received any support from AICc weighting. Model averaging L∞ based on AICc weights resulted in an estimated L∞ of 945 mm, an estimate 6 mm shorter than the that predicted by the von Bertalanffy model. In contrast to AICc values, BIC values indicated that the Gompertz model fit the data better than the Schnute–Richards model. Both Gompertz and Schnute–Richards models fit the data better than the logistic and Schnute models according to BIC values, as was indicated by AICc values.

Table 4 Statistical measures of fit for growth models fit to raw Gulf Corvina age-length data bolstered by simulated values.

The von Bertalanffy growth model described the data best according to AICc and BIC values. However, AICc weighting indicated that the Schnute- Richards and Gompertz models had nearly equivalent fits. Note: K indicated the number of parameters in each model.

Model	K	AICc	ΔAICc	AICc weight	BIC	ΔBIC	
von Bertalanffy	4	18,678.72	0.00	0.53	18,700.20	0.00	
Schnute–Richards	6	18,679.65	0.94	0.33	18,711.87	11.67	
Gompertz	4	18,681.29	2.57	0.15	18,702.77	2.57	
Logistic	4	18,702.60	23.89	0	18,724.09	23.89	
Schnute	3a	19,891.72	1,213.01	0	19,907.84	1,207.64	
Notes.

a Three parameters were estimated by nonlinear least squares, but four additional parameters were manually inputted (maximum and minimum ages and lengths) for the Schnute model.

Froese and Binohlan’s empirical relationship

Froese and Binohlan’s empirical relationship between Lmax and L∞ predicted a L∞ of 1,006 mm from a maximum observed length of 1,013 mm. This estimate is larger than all estimates of L∞ derived from growth models fit to raw data (Table 1) and raw data bolstered by simulated values (Table 3).

Literature review

Results from our review of 24 sciaenid growth studies and citations are summarized in Table  5. Age and body length relationships in sciaenids were represented by the von Bertalanffy growth model in 20 of 24 (83%) of the studies we reviewed, as reported in Atlantic Croaker (Micropogonias undulatus), Black Drum (Pogonias chromis), Red Drum (Scianops ocellatus), Southern Kingfish (Menticirrhus americanus), Spotted Seatrout (Cynoscion nebulosus), Weakfish (Cynoscion regalis), and Whitemouth Croaker (Micropogonias furnieri). Notably, growth of the Totoaba (Totoaba macdonaldi), a sciaenid closely related to Gulf Corvina and also found in the Gulf of California, was modelled by the von Bertalanffy growth model. Growth of the Gulf Corvina was modelled using the von Bertalanffy growth model in two previous studies.

Table 5 Models used to describe growth in sciaenid fishes similar to and including the Gulf Corvina.

A review of 24 sciaenid growth studies indicated that sciaenid growth is most commonly modelled by the von Bertalanffy growth model (20 studies; 83% of studies reviewed).

Growth model selected	Genus and species	References	
Gompertz	Cynoscion nebulosus	Murphy & Taylor, 1994 (females only)	
Linear	Cynoscion nebulosus	Murphy & Taylor, 1994 (males only); Nieland, Thomas & Wilson, 2002	
Logistic	Cynoscion nebulosus	Dippold et al., 2016	
Schnute–Richards	Cynoscion othonopterus	Aragon-Noriega, 2014 (selected for two datasets) Schnute Cynoscion othonopterusMendivil-Mendoza et al., 2017	
von Bertalanffy	Cynoscion nebulosus	Rutherford, Thue & Buker, 1982; Maceina et al., 1987; Wieting, 1989; Cottrell, 1990	
	Cynoscion othonopterus	Gherard et al., 2013; Aragon-Noriega, 2014 (selected for two datasets)	
	Cynoscion regalis	Lowerre-Barbieri, Chittenden & Barbieri, 1995; Hatch & Jiao, 2016; White, 2017	
	Menticirrhus americanus	Clardy et al., 2014	
	Micropogonias funieri	Manickchand-Heileman & Kenny, 1990; Santos et al., 2017	
	Micropogonias undulates	Barger, 1985; Barbieri, Chittenden Jr & Jones, 1993; Franco, 2014	
	Pogonias chromis	Murphy & Taylor, 1989	
	Sciaenops ocellatus	Beckman, Fitzhugh & Wilson, 1988; Murphy & Taylor, 1990; Ross, Stevens & Vaughan, 1995	
	Totoaba macdonaldi	Rodriguez & Hammann, 1997	

Other growth models have been used to describe age and body length relationships in sciaenids in five of 24 (21%) the studies we reviewed. The Gompertz model was used to model growth in female Spotted Seatrout. A linear model was used to model growth in male Spotted Seatrout, although it was noted that the linear growth pattern may have been due to lack of sampling of large and old individuals. Multi-model approaches similar to this study were employed three times (13%). One study on the Gulf Corvina selected the von Bertalanffy growth model for two datasets, and the Schnute–Richards model for two other datasets. As such, this study was counted in as one of the 20 studies that used the von Bertalanffy growth model to model sciaenid growth, and as one of the five studies that employed other growth models. The most recent age and growth study on Gulf Corvina selected the Schnute model, but did not describe a biphasic growth pattern. The other study that employed multiple models fit them to Spotted Seatrout age–length data and found the most statistical support for the three-parameter logistic model.

Model selection

Synthesis of the above considerations and assessments led to the selection of the von Bertalanffy growth model as the best model to represent Gulf Corvina growth. Of models fit in this study, the von Bertalanffy growth model had the 4th best statistical fit to raw data (AICc = 8,813.66, Δ AICc = 53.84, AICc weight = 0; BIC = 8,832.08, Δ BIC = 44.66; Table 2) and the best statistical fit to raw data bolstered by simulated values (AICc = 18,678.72, Δ AICc = 0. AICc weight = 0.53; BIC = 18,700.20, Δ BIC = 0; Table 4). Of models fit in this study, the von Bertalanffy growth model produced an L∞ closest to the L∞ predicted by Froese and Binohlan’s empirical relationship of 1,006 mm (raw data: 916.05 mm; raw data bolstered by simulated values: 951.30 mm; Tables 1 and 3). Further, the von Bertalanffy growth model was used to represent sciaenid growth in 83% of studies reviewed.

Simulations with a per-recruit model

The current value of the exploitation rate of old adults of Gulf Corvina EOA that we estimated is ca. twice larger than the value of EOA at which the YPR of Gulf Corvina reaches a maximum, regardless of the growth model employed for simulations with the per-recruit model (Appendix S1). However, the current FNSSBR of Gulf Corvina predicted when using the Schnute–Richards growth model fit to raw data only (0.60) is noticeably greater than the current FNSSBR of Gulf Corvina predicted when using all the other growth models (0.42–0.53) (Fig. 6A). The value of fraction of natural SSBR that causes population collapse typically is in the range of 0.1–0.6 and lower for short-lived fish species such as Gulf Corvina (Myers, Bowen & Barrowman, 1999; Grüss, Kaplan & Robinson, 2014). Thus, the per-recruit model employing Schnute–Richards growth model predicts the Gulf Corvina stock to be in a much healthier state than the per-recruit models employing other growth models. The current YPR over Y PRmax of Gulf Corvina predicted when using the Schnute–Richards growth model fit to raw data only (0.80) is also greater than the current YPR over Y PRmax of Gulf Corvina predicted when using all the other growth models (0.70–0.74) (Fig. 6B).

Figure 6 Consequences of using different growth models on (A) the current female SSBR over natural SSBR and (B) current YPR over Y PRmax of Gulf Corvina (Cynoscion othonopterus).

Per-recruit models employing the S-R growth model showed the stock to be in a healthier state than per-recruit models employing other growth models. SSBR, spawning-stock-biomass-per-recruit; YPR, yield-per-recruit; Gherard model, von Bertalanffy model developed in Gherard et al. (2013); VB model, von Bertalanffy model fit to raw data in the present study; VB bolstered model, von Bertalanffy model fit to raw data bolstered by simulation values in this study; S-R model, Schnute–Richards model fit to raw data in the present study; S-R bolstered model, Schnute–Richards model fit to raw data bolstered by simulation values in this study.

Discussion

This study illustrates the pitfalls of using statistical considerations alone when selecting a growth model for a vulnerable and highly-exploited species, due to the high likelihood of a biased distribution of samples. The combination of highly efficient, size-selective gear and high fishing effort have altered the age structure of the Gulf Corvina population (Ortiz et al., 2016), making sufficient representation of each size and age class difficult (Erisman et al., 2014). The influence of the lack of large and old fish in the dataset used for this study is clear, as Froese & Binohlan’s (2000) empirical relationship predicted an L∞ that was 89.95–275.09 mm greater than the L∞ predicted by growth modeled fit to raw data. This predicted L∞ from Froese & Binohlan’s (2000) empirical relationship of 1,006 mm was identical to the L∞ estimated by Gherard et al. (2013) with the von Bertalanffy growth model. Growth patterns (Figs. 4 and 5) and parameter estimates (Tables 1 and 3) were far less variable for models fit to data where sample size was equal at age, compared to models fit to raw data alone.

Unfortunately, heavy exploitation of Gulf Corvina makes the use of biased data in age and growth studies an unavoidable reality. Length data collected from the continuous monitoring of the fishery (Erisman et al., 2015; Ortiz et al., 2016) suggest that Gulf Corvina may be able to grow longer than their maximum reported length of 1,013 mm and live longer than their maximum reported age of 9 years, but heavy exploitation (exploitation rate of 0.825 year−1 for Gulf Corvina five years-old and older; Erisman et al. (2014) prevents them from doing so. Thus, despite our best efforts, we were unable to sufficiently represent large and old fish in our dataset. Further, while we made a concerted effort to sample small individuals, our data set would have been improved if we were able to collect more. This led to our decision to employ our simulation exercise to understand how the biases in our data affected our results. Similarly biased data have been the only data available for age and growth studies with Gulf Corvina, and previous studies have taken markedly different approaches to dealing with its limitations. Gherard et al. (2013) chose to use the inflexible and widely comparable von Bertalanffy growth model while acknowledging the data’s limitations and caveating results accordingly. Alternatively, Aragon-Noriega (2014) and Mendivil-Mendoza et al. (2017) employed a multi-model approach that leaned exclusively on statistics. Their statistical procedures led to the selection of the flexible Schnute–Richards and Schnute models to describe Gulf Corvina growth. Despite reporting different growth patterns depending on which dataset was used, Aragon-Noriega (2014) did not acknowledge the limitations of fishery-dependent data and concluded that Gulf Corvina grew in a biphasic pattern.

The same suite of models employed by Aragon-Noriega (2014) were fit to our data, and statistical measures of fit similarly supported the Schnute–Richards model as the best model for Gulf Corvina. Further, a biphasic growth pattern was described by the model, as it did in Aragon-Noriega’s (2014) study. However, our review of 24 sciaenid growth studies indicated that only one (4%) study used the Schnute–Richards model to describe sciaenid growth (Aragon-Noriega, 2014; Mendivil-Mendoza et al., 2017), and it was only this study that described a biphasic growth pattern. The biological implausibility of this growth pattern was further supported by the distance between Froese and Binohlan’s predicted L∞ and the L∞ estimated by the Schnute–Richards model (1,006 vs. 730.91 mm, i.e., a 275.09 mm difference). Finally, simulating an ideal sampling scenario where each age class was equally represented revealed that this biphasic growth pattern was due to bimodal distribution of samples and a lack of large, old fish. Both the parameter estimates and growth pattern changed substantially when simulated data was added to raw data so that sample size was equal for each age (Fig. 5 and Table 3). The Schnute–Richards model is flexible by design, and is, therefore, not suited for use with datasets that do not sufficiently represent each size and age class. Thus, by integrating the results of our statistical measures of fit, literature review, and simple indicators of biased data, we selected the von Bertalanffy growth model as the best model to represent Gulf Corvina growth.

The results of our study reinforce the well-established, but often forgotten, principle that each size and age class must be sufficiently represented for growth modelling to produce biologically reasonable results (Cailliet et al., 1986; Cailliet & Tanaka, 1990; Francis & Francis, 1992; Cailliet & Goldman, 2004). Ensuring sufficient representation may be difficult for highly-exploited fishes, as exploitation alters the population structure of fishes by preferentially selecting for large and old fish individuals (Berkeley et al., 2004; Mason, 1998). Similar challenges are faced when studying growth for vulnerable fishes or in data-poor fisheries, where there may not be resources available for extensive fishery-independent sampling or fish are scarce in general. Despite difficulty, ensuring sufficient representation of each size and age class should be a priority. The distinction between sampling to sufficiently represent each size and age class and sampling to represent population structure is an important one to make, as sampling to represent population structure should not be a goal of age and growth studies due to the bias created by the natural scarcity of large and old fish. As such, the simple simulation of an ideal sampling scenario with an equal number of samples at each age was not intended to represent the population structure of Gulf Corvina or to reflect the relative probability of obtaining samples of particular size and age classes in the real world.

Life-history parameters such as those estimated in growth models are influential in assessments for vulnerable and data-poor species (Fournier et al., 1990; Dulvy et al., 2004; Froese, 2004; Honey, Moxley & Fujita, 2010; Hordyk et al., 2016). These types of assessments rely on age–length data to determine vulnerability and overfishing, and problems emerge when all size and age classes are not sufficiently represented. The average maximum length (L∞) is underestimated and the growth rate (K) is overestimated when large and old fish are absent. Accordingly, a short generation time and lower levels of mortality are estimated, conferring more resilience to exploitation that the population possesses (Campana, 2001; Goldman et al., 2012; Harry, 2017). This idea was demonstrated with simulations with a per-recruit model, where the per recruit model using the Schnute–Richards growth model fit to raw data (which had the lowest L∞ in the present study) predicted Gulf Corvina reproductive capacity to be in a much healthier state than the per recruit models using other growth models (Fig. 6A). This false resiliency makes fishery management measures less effective (Campana, 2001; Cailliet & Andrews, 2008; Goldman et al., 2012), and may be present in Gulf Corvina assessments, as length and age truncation in the catch has increased progressively since biological monitoring of the fishery began in 1997 (Erisman et al., 2014). Given this length and age truncation, published growth models reported for this species may not be representing biology but rather the influence of exploitation.

Our results have implications for estimating growth within a stock assessment. Piner, Lee & Maunder (2016) documented an increase in precision in parameter estimates, ability to account for selectivity, and ability to incorporate multiple data sources when growth was estimated within a stock assessment. However, the influence of sample distribution on model output should be carefully examined if this approach is to be taken. So-called haphazard sampling strategies that ensure that all age and size classes are represented (e.g., Wells et al., 2013) make growth estimation within a stock assessment model more difficult, though are necessary if the density of samples at a particular age is driving model fit or preventing accurate estimation of L∞. Precision may be improved, but care must be taken to ensure that precision is being improved around biological reality.

We found that Gulf Corvina exhibit a high degree of variation in length-at-age, a pattern that is common among coastal fishes in the Gulf of California and other regions of the eastern Pacific characterized by significant annual variations in precipitation, ocean temperatures, and productivity in response to climate forcing that are known to influence growth rates in marine fishes (e.g., El Niño Southern Oscillation, ENSO; (Wells et al., 2006; Williams et al., 2007; Black, 2009). Specifically, growth rate is higher in Gulf Corvina during El Niño years, mainly in association with increased sea surface temperatures in the region (Reed, 2017). ENSO has been shown to affect fish growth in other areas, such as the waters of New Caledonia (Lehodey & Grandperrin, 1996), New Zealand (Gillanders et al., 2012), and north-western Australia (Ong et al., 2015; Ong et al., 2016). As the present study was conducted over multiple years, it is reasonable to assume that variations in ENSO over the study period (i.e., a succession of El Niño/La Niña events) could have affected the fish harvested for this study, explaining the observed variation in length-at-age. These effects may affect estimates of growth derived with growth models, but in order to identify these effects with confidence, complete sampling must be conducted with this purpose in mind. Another explanation for length-at-age variation could be measurement error (Neilson, 1992; Campana, 2001). Most of the fish in this study were harvested during their spawning season, which is around the time when they form new annuli. Therefore, it would be reasonable to conservatively assume that the precision of this study is more or less one year of age. However, we have high confidence in our reading of these otoliths, as annuli are clearly seen with minimal preparation (Fig. 1), and we excluded any samples for which there was a disagreement between readers.

Assessing the biological feasibility of growth model output, here accomplished with the use of simple indicators of biased data and literature review, is crucial for age and growth studies. Statistical measures of fit alone may not lead to the selection of a model that represents biological reality (Wang, Thomas & Somers, 1995; Cailliet et al., 2006; Araya & Cubillos, 2006). By integrating measures of statistical fit with results from the simple indicators and literature review, we concluded that the von Bertalanffy growth model best described the growth of Gulf Corvina and was most appropriate for the quality of available data. Though the Schnute–Richards model had the best statistical fit, it was not biologically reasonable, comparable between studies, or robust to biased data. Simple indicators such as those described in this paper should be used to reveal biases in data, and the use of flexible growth models such as the Schnute–Richards model to represent the growth of Gulf Corvina and similarly exploited fishes should be halted if biases are not accounted for.

Conclusions

Sample distribution influences growth model output, especially for flexible, statistically-driven models. Data used in growth modelling studies should be thoroughly examined for bias, as statistical measures of fit are insufficent as the sole criteria for selecting a model that reflects biological reality. Reflecting biological reality in growth models is critical for vulnerable fish and in data-poor fisheries, where age–length data are integral to assessing vulnerability and overfishing. In this case, the von Bertalanffy growth model represented biological reality best among the models tested. We warn against the production and use of growth models without recognizing biases in data given the serious implications for stock assessments and the management of vulnerable fish populations and data-poor fisheries.

Supplemental Information

Supplemental Information 1 Supplementary information

Click here for additional data file.

Supplemental Information 2 Alternative length-at-age models considered for Gulf Corvina (Cynoscion othonopterus) in our per-recruit model

aR = age of sexual maturity (2 years; Gherard et al., 2013) - aOA = age of transition from the young adult stage to the old adult stage (5 years).

Click here for additional data file.

Supplemental Information 3 Weigh-at-age model considered for Gulf Corvina (Cynoscion othonopterus) in our per-recruit model

aR = age of sexual maturity (2 years; Gherard et al., 2013) - aOA = age of transition from the young adult stage to the old adult stage (5 years).

Click here for additional data file.

Supplemental Information 4 Yield-per-recruit as a function of the exploitation rate of old adults for Gulf Corvina (Cynoscion othonopterus) (thick black curve), when alternative growth models are used

In each panel, the dashed-dotted black lines indicate the current exploitation rate of old adults of Gulf Corvina and the corresponding value of yield-per-recruit. Moreover, in each panel, the dashed grey lines indicate the exploitation rate of old adults of Gulf Corvina at which the yield-per-recruit of the species reaches a maximum and the corresponding value of yield-per-recruit.

Click here for additional data file.

Supplemental Information 5 Female spawning-stock-biomass-per-recruit (SSBR) over natural SSBR as a function of the exploitation rate of old adults for Gulf Corvina (Cynoscion othonopterus) (thick black curve)

In each panel, the dashed-dotted black lines indicate the current exploitation rate of old adults of Gulf Corvina and the corresponding value of yield-per-recruit. Moreover, in each panel, the dashed grey lines indicate the exploitation rate of old adults of Gulf Corvina at which the yield-per-recruit of the species reaches a maximum and the corresponding value of yield-per-recruit.

Click here for additional data file.

Table S1 95% confidence intervals for von Bertalanffy growth model parameters estimated from raw data

Click here for additional data file.

Table S2 95% confidence intervals for Gompertz growth model parameters estimated from raw data

Click here for additional data file.

Table S3 95% confidence intervals for logistic growth model parameters estimated from raw data

Click here for additional data file.

Table S4 95% confidence intervals for Schnute growth model parameters estimated from raw data

Click here for additional data file.

Table S5 95% confidence intervals for Schnute–Richards growth model parameters estimated from raw data

Click here for additional data file.

Supplemental Information 6 Code for analysis on raw data

Click here for additional data file.

Supplemental Information 7 Code for generating simulated values and analysis on raw data bolstered by simulated values

Click here for additional data file.

Supplemental Information 8 Code for creating histograms of age and length data

Click here for additional data file.

Supplemental Information 9 Ages and lengths for 749 Gulf Corvina collected from 2009–2013 in the Gulf of California, Mexico

Click here for additional data file.

Supplemental Information 10 Matlab code for estimating parameters in ypr and ssbr models

Click here for additional data file.

Supplemental Information 11 Matlab code to plot ypr and ssbr functions

Click here for additional data file.

Supplemental Information 12 Matlab code to plot per recruit curves

Click here for additional data file.

We acknowledge the Gulf of California Marine Program at Scripps Institution of Oceanography, project participants from El Centro para la Biodiversidad Marina y la Conservacíon and the fishers of the upper Gulf of California, Mexico for their contributions to data collection efforts. We thank Gregor Cailliet for his pre-submission comments on the manuscript, Chip Cotton, Brian Moe, and Grant Scholten for their feedback during data analysis, and Tyler Loughran for her help with formatting figures.

Additional Information and Declarations

Competing Interests

Author Contributions

Animal Ethics

Field Study Permissions

Data Availability

The authors declare there are no competing interests.

Derek G. Bolser conceived and designed the experiments, analyzed the data, prepared figures and/or tables, authored or reviewed drafts of the paper, approved the final draft.

Arnaud Grüss analyzed the data, prepared figures and/or tables, authored or reviewed drafts of the paper, approved the final draft.

Mark A. Lopez analyzed the data, approved the final draft.

Erin M. Reed performed the experiments, analyzed the data, prepared figures and/or tables, authored or reviewed drafts of the paper, approved the final draft.

Ismael Mascareñas-Osorio performed the experiments, analyzed the data, contributed reagents/materials/analysis tools, approved the final draft.

Brad E. Erisman conceived and designed the experiments, performed the experiments, analyzed the data, contributed reagents/materials/analysis tools, prepared figures and/or tables, authored or reviewed drafts of the paper, approved the final draft.

The following information was supplied relating to ethical approvals (i.e., approving body and any reference numbers):

Research protocol was approved under University of California San Diego IACUC ID no. S13240.

The following information was supplied relating to field study approvals (i.e., approving body and any reference numbers):

Data were collected under CONANP permit no. CNANP-00-007.

The following information was supplied regarding data availability:

The raw data and code are provided in the Supplemental Files.

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
