# Peer review of "The influence of sample distribution on growth model output for a highly-exploited marine fish, the Gulf Corvina (Cynoscion othonopterus)"

_PeerJ, doi:10.7717/peerj.5582_

## Round 0.1 · original submission · Major Revisions

Authors should underline the originality of the work and, at the same time, they should improve the focus of the aims of the manuscript, according to the reviewers' suggestions.

Reviewer 1 ·

Basic reporting

The paper is very well written. The Introduction is very clear, the background is carefully documented with relevant and well referenced literature. Perhaps the Introduction is too long (it seems a small review around this topic), but it could be very useful for readers.

Experimental design

The protocol used to fit the different growth models and to compare them is also sound, very well explained and robust. The parallel usage of simulation and experimental data is the best way to evaluate the different aspects and drawbacks of each model.

Validity of the findings

The results and the related discussion are also clear and coherent. I didn’t find weakness or inconsistencies in the approach. However, the only weakness of the paper is represented by the lack of the originality of the finding and of the related “message” of the paper. As the authors honestly declared in the abstract, “Our results serve as a reminder of the importance of complete sampling of all size and age classes” and, unfortunately, it is hard to find more than a reminder in the contribute of this paper to the relative field of work. Other sentence that definitively allows assessing the relevance of the paper are present in the discussion. Ex. Lines 346-348: “Similarly biased fishery-dependent data have been the only data available for age and growth studies with Gulf Corvina, and previous studies have taken markedly different approaches to dealing with its limitations” or Lines 375-378: “The results of our study reinforce the well-established principle that each size and age 376 class must be sufficiently represented for growth modelling to produce biologically reasonable 377 results (Cailliet et al., 1986; Cailliet and Tanaka, 1990; Francis and Francis, 1992; Cailliet and 378 Goldman, 2004)”. Yes, I strongly agree with the authors, but then why the readers should be interested in this paper? I’m sorry for this comment.

Additional comments

What I would like to suggest to the authors is one between two options: 1) rewrite some parts of the paper to stress the originality of the finding. Perhaps the authors could explore the cascade issues linked to the usage of wrong growth estimates in the stock assessment models? 2) transform the paper into a small review around this topic. At the present stage the paper is more interesting as a source of reference and as an historical reconstruction of this research framework.

Reviewer 2 ·

Basic reporting

The text is overwritten -especially in the Introduction and the Discusssion should be better organized. See general comments

Experimental design

The issues re; interpretation of groeth models in fish--Are we seeking to ek
lucidate the biology or are seeking to provide a description of the sample population and thus assess the impacts of fishing are important issues and should be spelt out more explicitly

Validity of the findings

The data presented are fine but would be greatly improved by some fishery independent sampling of small individuals

Additional comments

Bolser et al Review
General
At first sight, this paper deals with a number of issues that appear repetitively in the field of fisheries management, the treatment of which become technical and doctrinaire. However there are some important issues that are canvassed in this study and it emerges as something more than just another debate about the best way to model size at age data. As the authors point out under-representation of the maximum size and ages that the population members can achieve leads to underestimates of generation time and turnover rates. Agreed. They make the point if one does not ensure adequate representation of all age classes (read here, especially the large old fish) then “published growth models ---may not be representing biology but rather the influence of exploitation.” There are however likely to be a number of fisheries biologists, conservationists etc who would argue that measuring the influence of exploitation is exactly what is needed.
The Discussion that deals with this raises a number of important issues. Considerable variation occurs in the age structure of fish populations for a number of reasons; variation in recruitment success and the year class phenomenon, consistent selection of size and age classes by fisheries are the obvious ones. The authors commendably call for data sets that will result in the generation of demographic parameters that represent that true biological status of the population. This usually means capturing the influence of larger older members on generation times and mortality rates. The importance of this depends on the rate at which parameters are modified in a Darwinian sense—the salmonid people suggest this can be rapid. A better understanding of biology of the fished species requires information on possible increases in initial growth rates and reduction of the age at maturity. Selection imposed by the removal of the most fecund members of the population might be driving the age at maturity downwards. Is adding additional older individuals when analyzing such a population going to be realistic? We don’t know, but the point is worth debating. I think the authors handle these aspects fairly well but they need to be clearer about the alternative views as they have different consequences for fisheries management—and will likely attract some rebuttals.
If you want biological realism, you are not going to get it without fisher- independent sampling. In the present case, the focus should be on sub-adults, especially those that have not recruited into the fishery. There are two reasons for this. If there is going to be a response to fishing then this is where you will find it. Secondly getting better representation of the very early age classes is the best way to get realistic values of K (see below).
Government regulations usually require evaluations of the sustainability of fished stocks. This places a heavy responsibility on the development of fisheries models that estimate rate processes especially growth and mortality by methods where you don’t have actual age estimates. This study through comprehensive analysis of otoliths gets us past that point. However to do this properly you also require fishery-independent data, not simply balancing up the year classes by age-class information derived from the fishery (see line 106).

Specific

Lines 40-45. Introduction. This is overwritten and over-referenced. The intro could start at line 45.
Figs 4 & 5 Line 72. Are the authors happy with the use of K as a rate? K is a reciprocal of
time with units of time-1 (not length.time-1 as would be expected for a growth rate). Rather,
K varies with the age at which the growth trajectory reaches (or would reach) asymptotic
length L∞, and is a measure of the curvature of a growth trajectory. This point has been debated but much of the fisheries literature uses K as a measure of growth. So be it. However, there is an empirical point that is clear from Figs 4 and 5. The value of k is very sensitive to the distribution of size at age estimates of younger fish. If you miss these then the backward projection of the growth curve gives untenable estimates of to. Inclusion of smaller individuals usually increases K as does constraining the curve to a realistic settlement size. Many people don’t want to do this but if you don’t it is incumbent to have better representation of the younger age classes.
Lines 53, 77,381,392. Data-poor fisheries. Suggesting data poor fisheries will deliver poor service because they may use inaccurate estimates of L∞, K and other parameters is naïve. The problem with using data poor-estimates (especially in the Prince Hordyk papers) is that high levels of localized spatial variation occur in fish demographic parameters and are not taken into account. The consequent fisheries tools are used in areas where the demographic parameters are more biased than anything discussed in this paper. There are also major episodes of temporal variation due to climate forcing that reduce the value of estimates from data-poor models many of which use related species as proxies. The data sets in Figs 4 and 5 with high variances in size for each age class must reflect some of this. It is necessary to specify the spatial location of the sampling. With respect to temporal variation, the otoliths of this species would certainly allow the type of back calculation used by Black to take the impact of location-specific climatic variation (ENSO) into account.
Figs 4 & 5. The distribution of values for year classes 2 and 3 suggests the authors would do well to read Mulligan TJ, Leaman BM (1992) Length-at-age analysis: Can you get what you see? Can J Fish Aquat Sci 49:632–643.
There is now increasing evidence that populations at any given time may contain rapid and slow growing cohorts. Fast growing individuals will be very large at intermediate ages but do not survive. The older individuals may comprise members of the slow growing cohort. Back calculation to check on the growth rates with an age classes can sort this out.

In general the authors need a more explicit treatment of the alternatives—Do you want accurate age parameters in order to present a biologically realistic description of the population (achieved by ensuring better representation of each age class) or do you want models that reflect the impact of the fishery. At present, these tend to be buried in the Discussion. They need a clearer treatment as alternatives as they lead on to some important issues, the critical need for fishery-independent sampling, the need to get younger individuals for better estimates of K. Moreover, while they argue that we should be getter parameters that are biologically realistic, the selection imposed by fishing modifying age and reproductive schedules may be important aspects of the biology of the species in areas where fishing occurs.
They also need a better treatment of data-poor fisheries. Reading the discussion one gets the impression that the main problem is that they can generate inaccurate parameter estimates. This is not the case. What compromises data-poor estimates (even the better ones that don’t use other species as proxies) is that they ignore localized spatial variation and the temporal climate-driven impacts on growth rates. As the authors are seeking to make a general case for demographic sampling, they need to be more up front about the distorting impacts of spatial and temporal variation. My reading of the ms tells me that they have raised some interesting issues but need to improve the platform on which they are presented.

---

## Round 0.2 · accepted · Accept

Dear Authors,
I am pleased to confirm that your paper has been accepted for publication in PeerJ. Thank you for your efforts to improve your paper and for submitting your work to this journal.

# Reviewer 1 ·

Basic reporting

I confirm that the paper is very well written and organized. The authors did a remarkable effort to improve some issues of the manuscript. I am satisfied of this new version and of the replies to my comments. I think the paper is now worth of publication.

Experimental design

The experimental design is sound, well presented and support by data.

Validity of the findings

The findings are valid and useful to promote better researches in this framework

Additional comments

I appreciated the author replies and I agree with them.

Reviewer 2 ·

Basic reporting

The Introduction and Discussion are now more succinct and readable.

Experimental design

No further criticisms

Validity of the findings

Worthy of publication in the fisheries literature.

Additional comments

My congratulations to the authors on a detailed and balanced response to the critiques outlined in my review.

My support for the value of this ms is that the literature is now replete with examples that fit various growth models to data as an end in themselves and do not explore basic biological properties that may modify or drive model parameters in unexpected ways.